# A Low-Cost Sensing Solution for SHM, Exploiting a Dedicated Approach for Signal Recognition

**DOI:** 10.3390/s24124023

**Published:** 2024-06-20

**Authors:** Bruno Andò, Danilo Greco, Giacomo Navarra, Francesco Lo Iacono

**Affiliations:** 1Department of Electrical Electronic and Computer Engineering (DIEEI), University of Catania, 95123 Catania, Italy; bruno.ando@unict.it; 2Department of Engineering and Architecture, Kore University of Enna, 94100 Enna, Italy; giacomo.navarra@unikore.it (G.N.); francesco.loiacono@unikore.it (F.L.I.)

**Keywords:** structural health monitoring, sensing system, embedded architecture, time–frequency analysis, signals classification

## Abstract

Health assessment and preventive maintenance of structures are mandatory to predict injuries and to schedule required interventions, especially in seismic areas. Structural health monitoring aims to provide a robust and effective approach to obtaining valuable information on structural conditions of buildings and civil infrastructures, in conjunction with methodologies for the identification and, sometimes, localization of potential risks. In this paper a low-cost solution for structural health monitoring is proposed, exploiting a customized embedded system for the acquisition and storing of measurement signals. Experimental surveys for the assessment of the sensing node have also been performed. The obtained results confirmed the expected performances, especially in terms of resolution in acceleration and tilt measurement, which are 0.55 mg and 0.020°, respectively. Moreover, we used a dedicated algorithm for the classification of recorded signals in the following three classes: noise floor (being mainly related to intrinsic noise of the sensing system), exogenous sources (not correlated to the dynamic behavior of the structure), and structural responses (the response of the structure to external stimuli, such as seismic events, artificially forced and/or environmental solicitations). The latter is of main interest for the investigation of structures’ health, while other signals need to be recognized and filtered out. The algorithm, which has been tested against real data, demonstrates relevant features in performing the above-mentioned classification task.

## 1. Introduction

The continuous monitoring of buildings and structures is strategic, particularly with the aim of health assessment and preventive maintenance planning, especially in seismic areas [1]. Actually, the possibility of identifying potential risks associated with aging, as well natural hazards and environmental factors, is essential for disaster prevention [2].

Structural health monitoring (SHM) aims to provide a robust and effective approach to obtaining valuable information on structural conditions and supporting methodologies for the identification and, under specific conditions, localization of potential damages. Comprehensive reviews on SHM are available in [3,4,5,6], while their specific use in bridge monitoring is addressed in [7]. SHM can be implemented at either a global or local level [1]. Global monitoring is very convenient as it allows for assessing the overall behavior of the entire structure by measuring dynamic responses through the use of sensors (e.g., accelerometers or tiltmeters). On a different scale, the analysis of local cracks or fatigues requires other techniques like nondestructive testing (NDT).

SHM can be approached through two main strategies: discrete monitoring and continuous monitoring of health-related quantities. Discrete monitoring involves highly accurate measurements, typically sparse in time and performed only at specific locations. This approach lacks both time-continuous and spatially dense monitoring due to its requirement for accurate and costly instrumentation [8,9]. Conversely, the use of low-cost sensing nodes, often based on micro-electromechanical sensors (MEMS), can be conveniently adopted for the implementation of solutions allowing for the continuous monitoring of structural health. The development of a multi-sensor node for applications in the field of structural monitoring is addressed in [10], along with its assessment by real seismic signals.

Although these systems will show lower accuracy with respect to high-cost instrumentation, they offer the possibility of realizing sensor networks for time-continuous and spatially dense monitoring. This approach is valuable for the implementation of early warning systems (EWSs), allowing for the timely detection of anomalous structural behaviors [11,12]. It clearly emerges that EWSs are hence strategic in implementing monitoring tasks in many contexts, such as schools, hospitals, public or private buildings, and civil infrastructures (e.g., bridges). The commonly adopted methodology to fix the number and the location of sensing nodes often relies on the knowledge of reliable structural models, which could allow for optimizing the sensor network and identifying the most relevant monitoring locations [1]. Also, introducing redundancy in the number and location of monitoring nodes allows for improving the system’s information reliability. Optimizing data transmission strategies is also strategic for wireless sensor networks in general and, particularly, for EWSs involving the use of dense network architectures [13].

It is important to bear in mind that data provided by monitoring systems must be appropriately processed to extract relevant features, which are crucial for feeding dedicated algorithms aimed at assessing structural health and identifying the nature and potential locations of damage [14]. To such an aim, different approaches can be adopted. As an example, in [15], signal decomposition implemented through discrete wavelet transform (DWT) is proposed as a suitable approach to filtering out exogenous sources coming from the environment while highlighting main seismic sources. The development of a low-cost multi-sensor node, compliant with the early warning system for SHM and aimed to measure inertial vibrations and tilt, is reported in [16]. Also, this work presents a dedicated signal processing to assess the sensing node performance in the time and frequency domains. In [17], a continuous wavelet transform (CWT) operator has been introduced with the aim of analyzing the time–frequency content of signals provided by the sensing node investigated in [16]. In particular, cross-correlation and coherence operators have been used to validate the system performances against reference instrumentation.

This paper represents a follow-up with respect to the work presented in [15,16,17]. In particular, the aim of this work is twofold: a deep assessment of the multi-sensor node performance [16] and the development of a dedicated methodology to discriminate among different kinds of signals: (i) noise floor (NF), being mainly related to the intrinsic noise of the sensing system; (ii) exogenous sources (ES), not correlated to the dynamic behavior of the structure; and (iii) structural responses (SR), i.e., the response of the structure to external stimuli, such as seismic events and artificially forced and/or environmental solicitations. The latter is of main interest for the investigation of structures’ health, while other signals need to be recognized and filtered out.

Main novelties introduced by this paper concern the following:More detailed device assessment is provided by highlighting its behavior under different kinds of input stimulations.The use of dedicated key performance indexes aims to compare the behavior of the sensor node with respect to the ground truth. In particular, in addition to the traditional root-mean-square error, a time-correlation index and a Wavelet-correlation index have been introduced;The dedicated methodology for the classification of different signals is provided by the sensing system, using a threshold approach supported by the receiving operating curve (ROC) theory.

The main outcomes of the proposed approach are related to the following:There is a possibility to adopt an embedded sensing node for the implementation of continuous SHM; actually, the low-cost feature of this MEMS-based device, compared with highly accurate instrumentation usually adopted to perform discrete monitoring, enables the development of distributed EWSs.The use of CWT-based operators, with respect to discrete wavelet transform, shows a more accurate frequency sampling, thus providing a more reliable analysis of transients in the output signal.The methodology is implemented to separate exogenous dynamics from signals of interest (i.e., structural response), which are potentially useful to feed paradigms aimed at extracting health-related quantities and, consequently, tracking short- and long-term behaviors of the monitored structure. It must be considered that the last task is outside of the purpose of this work.

The activity presented through this work has been developed in the framework of the “HCH LowCost GeoEngineering Check” project, which aims to develop suitable methodologies for the continuous monitoring of buildings. The strategy adopted is based on the use of sensors and advanced tools for data processing, exploiting both standard instrumentation and low-cost sensing nodes, such as the one investigated in the following.

A brief description of the low-cost multi-sensor node architecture is reported in Section 2, along with the approach proposed for the classification of different classes of signals. Section 3 reports results related to the assessment of the sensing architecture and the validation of the classification algorithm. To achieve the former aim, the node is prodded by periodical stimuli with different frequencies. Moreover, tests using signals recorded during real events are also performed. During these experiments, reference instruments are used for the sake of comparison along with dedicated metrics. The classification algorithm, which has been tested against real data, demonstrates relevant features in performing the above-mentioned classification task.

## 2. Materials and Methods

### 2.1. Brief Description of the Multi-Sensor Node

Although the developed sensing platform has already been presented in [15,16], in the following, a brief description of its architecture and main functionalities is provided. Considering the addressed application and the need for a low-cost solution, the main specifications required for the tilt and acceleration measurements are as follows:Acceleration: range ±1.5 g, resolution 0.005 g;Tilt: range ±10°, resolution 0.02°;Frequency range: 0.5 ÷ 20 Hz.

As schematized in Figure 1, the sensing node consists of the following main parts [16,17]: The triaxial MEMS inclinometer SCL3300-D01-PCB by Murata, with working range set to ±10° full-scale and the tilt resolution of 0.0055°/LSB, with a 0.001 °/√Hz noise density;The MEMS triaxial accelerometer SCA3300-D01-PCB by Murata, set to an operating range of ±1.5 g and using a 70 Hz 1st-order low-pass filter;The embedded module RT1062 Teensy 4.0 series, based on a ARM^®^ Cortex^®^-M7 MPU at 600 MHz;The DS3234 real-time clock;The GPS MODULE—COPERNICUS II DIP;A micro-SD card for data storing;The HC-05 Bluetooth Bee Master Slave 2 in1 module;A lithium EEMB 3.7 V 2000 mAh 103,454 battery.

A real view of the device installed in a real environment is shown in Figure 2 [16]. The adopted sampling rate is 200 Hz, which is compliant with the addressed frequency range. The node can be connected to a PC through a USB interface and/or a Bluetooth protocol. Also, a graphical user interface (GUI) has been realized [16], allowing for a complete management of the node, including data transmission and visualization. Actually, a full set of commands are implemented in order to control and set the node operation. In particular, the implemented working modes allow for continuous or event-triggered data storing/transmission. In the latter case, “activation thresholds” are settable by the GUI.

### 2.2. The Classification Algorithm

The main aim of the developed algorithm is the detection and classification of different classes of signals: noise floor (NF), exogenous sources (ES), and structural responses (SR). In the following, the proposed classification approach is described, while the results obtained by its experimental assessment are given in Section 3.2. 

The first task accomplished toward the development of the classification methodology has been the collection of a dedicated dataset, including the following kinds of signals:NF recorded by the sensor node.ES acquired by exposing the sensor node to environment-related solicitation, such as human walking, home appliances, and pulse-like dynamics.SR whose strength must be compliant with the target sensitivity of the developed low-cost node.Since during the experimental survey, it was not possible to observe SR generated by natural events showing strength compliant with the sensor node specifications, signals recorded by a reference accelerometer positioned close to the sensor platform were used. Such signals were selected by considering the occurrence of low-strength seismic events. Dedicated post-processing was then implemented, with the aim of adapting the strength of acquired signals to values compliant with the characteristics of the sensor node.

Typical examples of the acceleration trends for the case of the above-mentioned signals are reported in Figure 3, along with their frequency contents. In particular, the CWT operator has been used to investigate the time–frequency characteristics of considered signals. 

Table 1 reports the number of patterns for each class of the collected dataset. Each pattern consists of a time window of 10 s of the acceleration module. The dataset is divided into two parts: 100 patterns for each class to be used as the test dataset, while the remaining patterns (setting dataset) are used for the identification of the classification model.

To develop a feature-based classification methodology, the root-means-square (RMS) value of the acceleration module, calculated across the 10 s time window, is used. This can be considered as a key feature conveying a strategic piece of information related to the signal strength in the considered time interval. Figure 4 shows the distribution of RMS values for the whole set of considered patterns. As can be observed by such distribution and the detail embedded in Figure 4, it is not possible to find a separation threshold allowing for a simple clustering of different kinds of signals. To better substantiate the above statement, the receiving operating characteristic (ROC) theory is used [18]. 

To such aim, the following sensitivity (*Se*) and specificity (*Sp*) quantities are used [18]:(1)$$Se=\frac{TP}{TP+FN}$$
(2)$$Sp=\frac{TN}{TN+FP}$$
where standard definitions of true positive (*TP*), true negative (*TN*), false positive (*FP*), and false negative (*FN*) are used [18].

Figure 5 shows trends of *Se* and *Sp* as a function of the considered threshold $\left(Th\right)$, for the two cases discriminating NF from other sources and SR from ES.

Optimal thresholds $\left(Th_{op}\right)$, allowing for the implementation of separation tasks, are estimated by minimizing the distance of *Se* and *Sp* from the condition of complete separation (*Se* = 1 and *Sp* = 1), through the following expression [18]:(3)$${\left(1-Se\left(Th_{op}\right)\right)}^{2}+{\left(1-Sp\left(Th_{op}\right)\right)}^{2}=\min\left({\left(1-Se\left(Th\right)\right)}^{2}+{\left(1-Sp\left(Th\right)\right)}^{2}\right)$$

In case condition (3) is achieved for different threshold values belonging to an interval, the mean value calculated across such interval is used as the optimal threshold.

As can be observed by the achieved results (Figure 5b), the classification task is worth improving. Further outcomes of this analysis are reported and discussed in Section 3.2.

On the basis of the above results, in order to extract other relevant features to effectively implement the classification task, a deeper investigation of the dataset has been performed. As evidenced by Figure 3, the three classes of signals show different characteristics in the frequency domain. In particular, NF shows a wide band energy spectrum, while ES and SR present opposite behaviors. The first one has most of its energy located at higher frequencies, whereas the latter is bounded at lower frequencies. The last statement is also supported by the literature, in which the typical frequency range of structural responses to natural/forced solicitation is mainly below 20 Hz, especially for tall buildings [19,20,21].

On the basis of the above findings, the whole dataset has been processed by a low-pass (LP) and a high-pass (HP) filter, both with a cutoff frequency of 20 Hz. RMS values of filtered data, calculated across the 10 s time window, namely, *RMS_LP_* and *RMS_HP_*, represent the new features to be used for implementing the classification task. 

As can be observed from Figure 6, the distribution of LP-filtered data demonstrates the possibility for clustering the NF class, while the same applies to the ES class in the case of HP data. 

As a first step, optimal thresholds, which allow for implementing the following separation tasks, have been estimated:
$Tr_{LP}$: to separate NF from ES and SR, by LP-filtered data;$Tr_{HP}$: to separate ES from NF and SR, by HP-filtered data.

Figure 7 shows *Se* and *Sp* values for the above tasks as a function of separation thresholds, *Tr_LP_* and *Tr_HP_*. The following optimal thresholds have been estimated: $Tr_{LP}=1.38\times 10^{-4}\;g$, $Tr_{HP}=2.69\times 10^{-4}\;g$.

Considering all the above statements, the following classification rules have been identified:(4)$$Class=\left\{\begin{array}{c} \begin{array}{cc} \mathrm{NF}: & RMS_{LP}\leq Tr_{LP} \end{array}\\ \begin{array}{cc} \mathrm{SR}: & \left\{\begin{array}{c} RMS_{LP}>Tr_{LP}\\ RMS_{HP}<Tr_{HP} \end{array}\right. \end{array}\\ \begin{array}{cc} \mathrm{ES}: & RMS_{HP}\geq Tr_{HP} \end{array} \end{array}\right.$$

The above rules are used to implement the algorithm shown in Figure 8, aimed at the real-time classification of data acquired by the sensing platform. Although compliant with its deployment in the adopted microcontroller platform, for the sake of convenience, at this stage, the classification algorithm is implemented in MATLAB R2022b.

The algorithm operates in time windows of 10 s, extracting the classification features from the time evolution of the acceleration module. Since the algorithm is expected to work continuously on real-time data sequences, a storing array is used, which is continuously updated with new data. As soon as a time increment of 1 s is achieved (condition checked by “shifting time check” routine), a new pattern containing the last 10 s of the time sequence is released and processed in order to extract features and estimate its class of belonging.

Summarizing, each pattern is processed as follows:A zero-order de-trend is accomplished, thus removing the effect of signal offset;The de-trended signal is then processed by LP and HP filters;*RMS_LP_* and *RMS_HP_* values are calculated;The set of rules (4) is applied to estimate the class of belonging for the considered pattern.

Results obtained by the above-presented classification algorithm are discussed in Section 3.2.

## 3. Results and Discussion

### 3.1. Assessment of the Low-Cost Sensing Node

In this section, results of experimental surveys performed to validate the developed sensing node are reported. For the sake of convenience, it must be considered that the device resolution has already been investigated in [16], leading to the results shown in Table 2.

In order to test the system response to acceleration stimuli, the experimental setup shown in Figure 9a is used, equipped with the APS 129 HF ELECTRO-SEIS^®^ long-stroke vibration exciter, available at the Experimental Dynamics Laboratory of the L.E.D.A. Research Institute at the University of Enna “Kore”, Enna, Italy [22]. The sensing platform is fixed to the moving platform of the shaker. The system, after a settling time, performs 10 reliable periods at the desired frequency and amplitude. The reference value is provided by a QA-700 accelerometer by Honeywell, Charlotte, NC, USA, whose main characteristics are as follows:Operating range: ±30 g;Bias: <8 mg;One-year composite repeatability: <1200 μg;Temperature sensitivity: <70 µg/°C;Intrinsic noise: <7 µg rms (0–10 Hz), 70 µg rms (10–500 Hz).

Characteristics of stimulation signals are reported in Table 3 in terms of their nominal frequency and amplitude values. Figure 9b shows the concatenation of 10 periods of the signal recorded by the sensing node for each considered frequency [17]. Results obtained by the wavelet analysis, shown in Figure 9c [17], as well as the discrete Fourier transforms shown in Figure 9d, confirm the compliance of the system response to the applied stimulus.

In order to quantify such performances, the following index has been defined:(5)$$\delta_{V}=100*\frac{1}{n}*\sum_{i=1}^{n}\left|\frac{V_{1}-V_{2}}{V_{1}}\right|$$
where *V* states for the amplitude, *A*, or the frequency, *f*, of the applied stimulus.

Moreover, the repeatability, assessing the system performances in the acceleration domain, is estimated as 3 times the standard deviation of peak values distribution. The results obtained for above defined indexes, under different operating conditions, are shown in Figure 10. The *δ_A_*, for the three axes, is limited to under 1% in most of the investigated frequencies, with the exception of a few cases where its value reaches 5%. The *δ_f_* values show that the frequency discrepancy is extremely low (less than 0.03%) for each solicitation frequency and for each axis. The repeatability has an upward trend with the frequency, which rises after 10 Hz, being, however, confined below 4.5% of the nominal values.

The next set of experiments is dedicated to testing the system response to dynamics imposed by using a controlled moving platform. The sensing node is hooked at the center of the vibrating platform, as shown in Figure 11. The vibrating platform has 6 degrees of freedom, with ±1.5 g horizontal acceleration range, ±1 g vertical acceleration range, and an RT3-S real-time digital control system, exploiting a 2 kHz control loop in position, velocity, and acceleration. A detailed description of the shaking table facility is reported in [23,24]. For the sake of comparison and validation, a dedicated reference system has been used, which consists of eight MEMS DC accelerometers (model 3711B1110G by PCB Piezotronics, Depew, NY, USA) placed in the actuator of the vibrating platform (two along the *X*-axis, two along the *Y*-axis, and four along the *Z*-axis), whose main specifications are as follows:Range ±10 g;Frequency range 0–1.0 kHz;Nonlinearity ≤1%;Transverse sensitivity ≤3%.

In particular, the response of the sensing node to the following signals is observed:

Frequency sweep test: an acceleration stimulus, ranging from 0.5 Hz to 20 Hz, applied along the *X*, *Y*, and *Z*-axes;Tilt test: a periodic tilt, ranging from −2.5° to 2.5°, applied for two different frequency values;Seismic test: a typical seismic signal.

The results obtained for the frequency sweep test are reported in Figure 12, which shows the output from both the reference system and the sensing platform. Figure 13 shows the wavelet power spectrum of the above signals [25]. In particular, to achieve this goal, the synchrosqueezing wavelet is used to narrow the frequency distribution in each time instant. The time–frequency analysis clearly demonstrates the coherence with the adopted frequency sweep stimulus.

The following indexes are used in order to assess the performance of the sensing platform:(6)$$\zeta =100*\sqrt{\frac{\sum_{i=1}^{n}{\left(V_{1}-V_{2}\right)}^{2}\;}{\sum_{i=1}^{n}{\left(V_{1}\right)}^{2}\;}}$$
(7)$$R_{V_{1},V_{2}}\left(l\right)=\left\{\begin{array}{c} \frac{\sum_{i=0}^{n-\left|l\right|-1}V_{1}\left(i\right)*V_{2}\left(i+l\right)}{n},l<0\\ \frac{\sum_{i=0}^{n-\left|l\right|-1}V_{1}\left(i\right)*V_{2}\left(i+l\right)}{n},l\geq 0 \end{array}\right.$$
(8)$$R_{norm}=\frac{R_{1,2}\left(0\right)}{\sqrt{R_{V_{1},V_{1}}\left(0\right)R_{V_{2},V_{2}}\left(0\right)}}$$
where*ζ* estimates the percentage error with respect to the nominal values on the whole signal;*V*_1_, *V*_2_ are signals recorded by the reference system and the sensing node, respectively;*n* is the number of samples;$R_{V_{1},V_{2}}\left(l\right)$ is the cross-correlation between the two signals, *V*_1_ and *V*_2_, as a function of the lag (*l*);*R_norm_* is the normalized cross-correlation between the reference and the acquired signals, defined in *l* = 0.

Moreover, the following index, $C_{norm}$, representing the normalized cross-correlation in the time–frequency domain, is used to assess the similarity of the time–frequency content of the sensor output against the reference signal:(9)$$C_{norm}=\frac{\sum_{xy}\left(\left(C_{1}\left(x,y\right)-\bar{C_{1}}\right)\left(C_{2}\left(x,y\right)-\bar{\bar{C_{2}}}\right)\right)}{\sqrt{\sum_{x,y}{\left(C_{1}\left(x,y\right)-\bar{C_{1}}\right)}^{2}\sum_{x,y}{\left(C_{2}\left(x,y\right)-\bar{C_{2}}\right)}^{2}}}$$
where*C*_1_ and *C*_2_ are two matrixes containing the wavelet coefficients;$\bar{C_{1}}$ and $\bar{C_{2}}$ are mean values of elements belonging to the above matrixes. 


The results obtained for performance indexes (6), (8), and (9) are reported in Table 4. High values of R_norm (close to 1.0) for the three axes prove the capability of the sensing platform to follow the dynamics of the solicitation in the whole investigated frequency range. Values of C_norm are over 0.9, which demonstrates the coherence in the frequency domain of the two signals. The ζ values provide a quantification of the difference in magnitude between the two signals, which, on average, is about 8.5%. As can be observed, the *Z* and *Y* axes show higher ζ values with respect to the *X*-axis, most probably due to the presence of noise superimposed to the output signal along these directions. Further investigations will be dedicated to identifying possible strategies aimed at reducing the effect of such influencing quantities on the system performance.

Concerning the tilt test, the vibrating platform generates a series of oscillations at fixed frequencies (0.2 Hz and 0.5 Hz) in the range of ±2.5°. The test is performed through the *X* and *Y* axes (as defined in Figure 11) of the sensor node, which is intended to measure a quasi-static tilt. Figure 14 shows the time series provided by both the reference system and the sensing platform. Performance indexes (6), (8), and (9) calculated for this test are reported in Table 5. As can be observed, estimated values for ζ are lower than 4%, and values of *R_norm_* and *C_norm_* are close to 1 for both considered axes. Such results demonstrate the capability of the sensor node to follow the imposed solicitation.

During the last test on the accelerometer, the vibrating platform was used to reproduce the ground motion recorded during a real earthquake. Particularly, the tree components of the Kobe earthquake, recorded in Takatori, Japan, on 16 January 1995, were used. This event was characterized by high horizontal and vertical ground motion (*M_w_* = 6.9). The test aimed to validate the sensing platform response to this kind of realistic complex solicitation (rich magnitude and frequency content). Figure 15 shows output signals of both the reference system and the sensing platform, for the three axes. From the wavelet analysis, shown in Figure 16, it is possible to observe that the energy of the wavelet in the *X* and *Y* axes is mainly located in the low-frequency range, while the response along the *Z*-axis shows relevant energy content for higher frequency. Table 6 reports performance indexes (6), (8), and (9) calculated for this test. Values of $R_{norm}$ are close to 1.0 for the three axes, thus validating the capability of the node to measure input with rich frequency content. Values obtained for the index ζ, along the *X* and *Y* axes are in line with the previous test, resulting in values below than 6%. Values of $C_{norm}$ for the *X* and *Y* axes confirm a suitable coherence in the time–frequency domain between the sensing node and the reference system, while highlighting lower coherence for the *Z*-axis. The worst performances observed for the *Z*-axis are most probably due to the presence of high-frequency noise along this direction.

### 3.2. Outcomes of the Classification Algorithm

Results obtained by applying the algorithm presented in Section 2.2 are shown in Figure 17 where blue symbols represent the expected class for each pattern and the red symbols the output of the classification procedure. For the sake of completeness, Figure 17a reports results obtained by the classification algorithm exploiting RMS values of the raw acceleration module and thresholds defined in Figure 5 where misclassifications of SR and ES patterns are clearly observable. Figure 17b shows results obtained by the algorithm, shown in Figure 8 and exploiting rules (4), which allows for overcoming the misclassification issue. To support this finding, Table 7 shows the confusion matrix obtained for the classification algorithm shown in Figure 8 for the setting and test datasets. As can be observed, the obtained results confirm the suitability of the classification methodology. 

## 4. Conclusions

In this paper, the development of a low-cost sensing platform for applications in the field of SHM is presented, along with a signal processing approach allowing for the analysis and classification of different kinds of dynamics recorded by the sensor node. The latter has shown suitable performances in both acceleration and tilt measurements.

In particular, tests performed by forcing periodic acceleration along the three axes, at 10 different frequencies in the operating range (0–20 Hz), revealed amplitude discrepancies of less than 1% in most of the frequencies tested, with the exception of a few cases where its value reached 5%.

Performing a frequency sweep from 0.5 to 20.0 Hz, the value of the correlation between the signal provided by the reference instrumentation and the signal recorded by the node was very close to 1, while the difference in magnitude between the two signals was, on average, about 8.5%.

The tilt test performed in the range of ±2.5° has shown a discrepancy lower than 4%. Finally, the test exploiting the reproduction of a real earthquake has demonstrated suitable performance of the sensing platform when solicited by realistic time series.

A dedicated approach for the classification of different kinds of signals recorded by the sensing node has been introduced and assessed. In particular, the algorithm developed allows us to separate three classes of signals, namely, noise floor, structural response, and exogenous signals. The algorithm has been tested by using a dedicated dataset, and the results obtained allow for affirming the excellent performance of the methodology developed.

Future activities will be dedicated to implementing the deployment of the preprocessing and classification algorithm in the embedded architecture of the sensor node. Moreover, a large experimental survey will be performed by recording long-lasting time series, thus extending the available dataset. If required, the procedure adopted to set classification thresholds will be iteratively applied in order to adapt the set of estimated rules to a wide set of real observations.

Moreover, efforts will be also dedicated to identifying possible strategies aimed at reducing the effect of influencing quantities on the system performances.

## Figures and Tables

**Figure 1 sensors-24-04023-f001:**
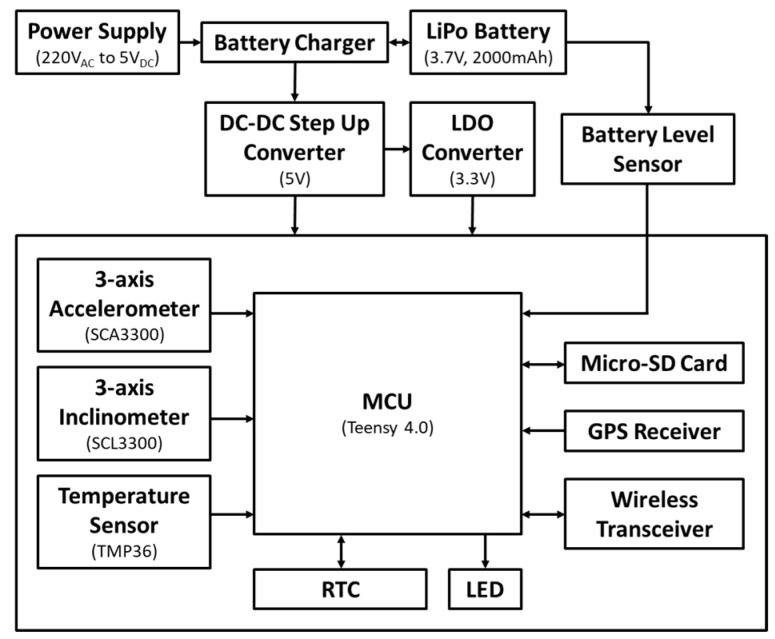
Schematization of the multi-sensor node [17].

**Figure 2 sensors-24-04023-f002:**
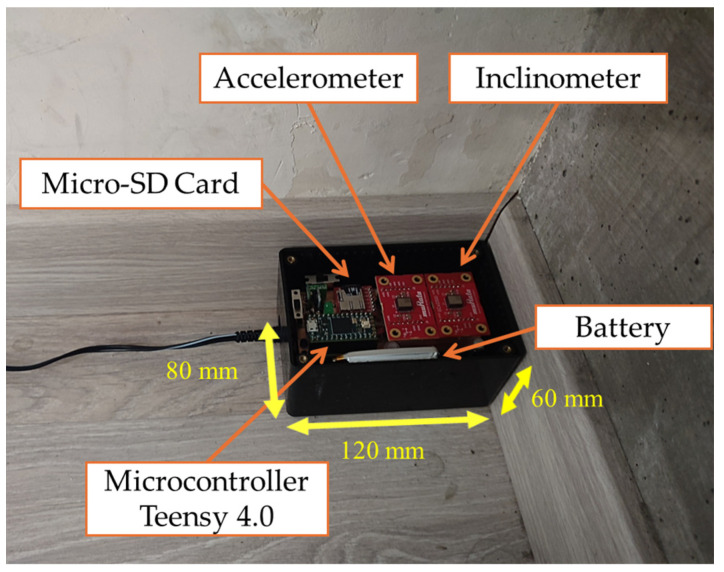
The sensor node [16] installed in a real environment.

**Figure 3 sensors-24-04023-f003:**
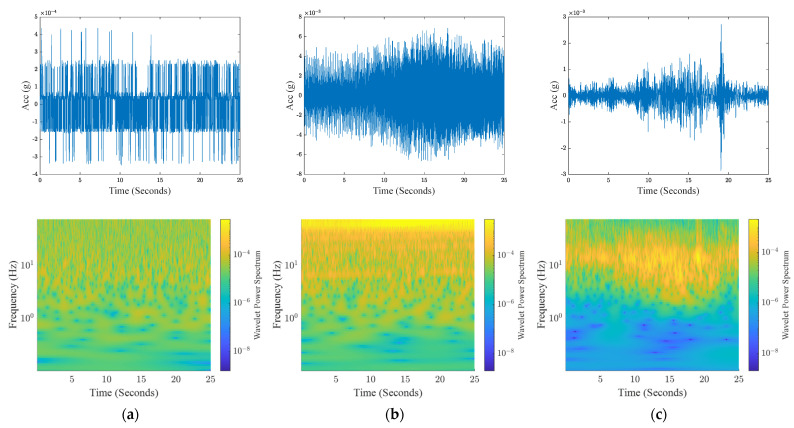
Time evolution of typical signals and their time–frequency representation: (**a**) NF, (**b**) ES, (**c**) SR.

**Figure 4 sensors-24-04023-f004:**
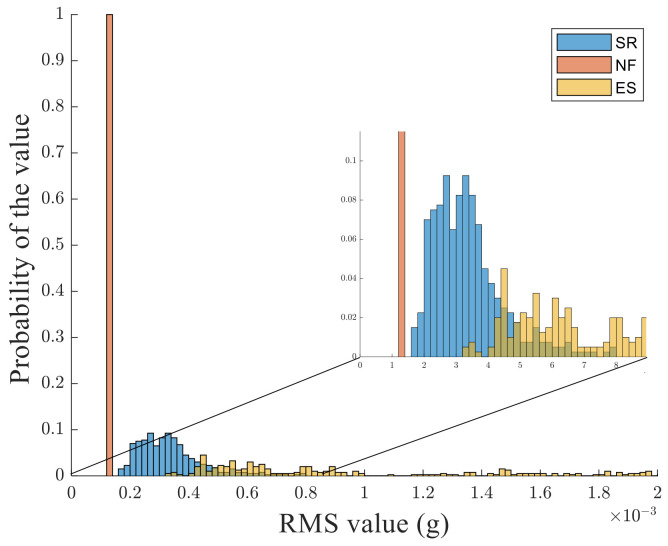
Distribution of RMS values for the whole set of considered patterns. The detailed view aims to show the superposition of patterns belonging to different classes.

**Figure 5 sensors-24-04023-f005:**
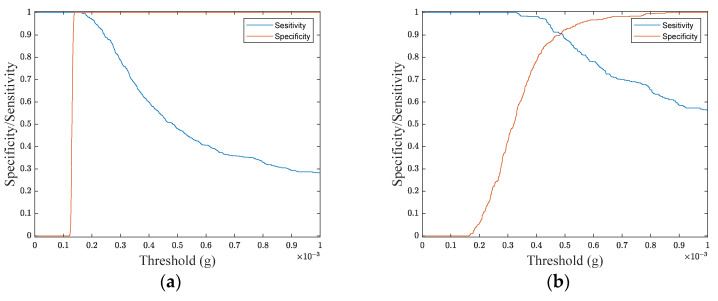
Sensitivity (*Se*) and specificity (*Sp*) values as a function of the considered threshold for the two cases discriminating (**a**) NF from other sources, (**b**) SR from ES.

**Figure 6 sensors-24-04023-f006:**
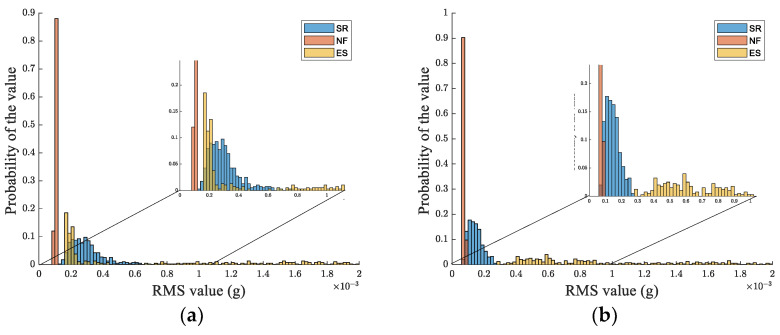
Distribution of RMS values for the whole set of considered patterns. The detailed view aims to show an overlap of patterns belonging to different classes. (**a**) Low-pass filtered signals; (**b**) high-pass filtered signals.

**Figure 7 sensors-24-04023-f007:**
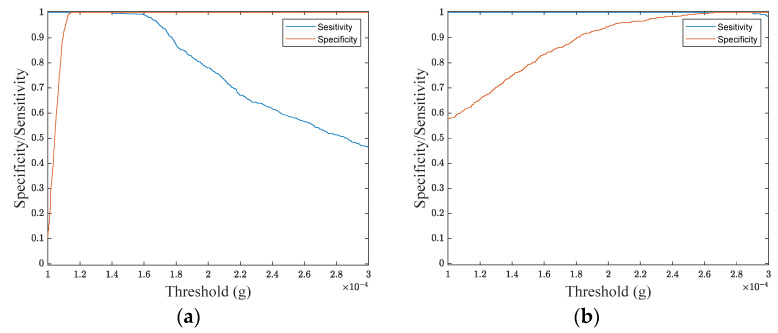
Sensitivity (*Se*) and specificity (*Sp*) values as a function of the considered threshold for the following tasks: (**a**) separation of NF from other sources by LP data and (**b**) separation of ES from other sources by HP data.

**Figure 8 sensors-24-04023-f008:**
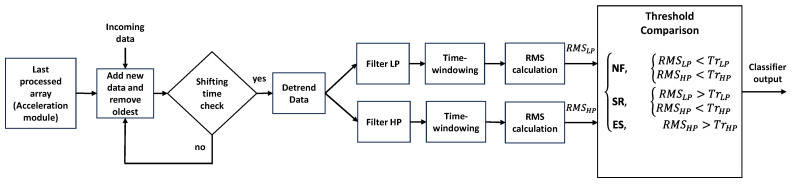
The real-time classification algorithm.

**Figure 9 sensors-24-04023-f009:**
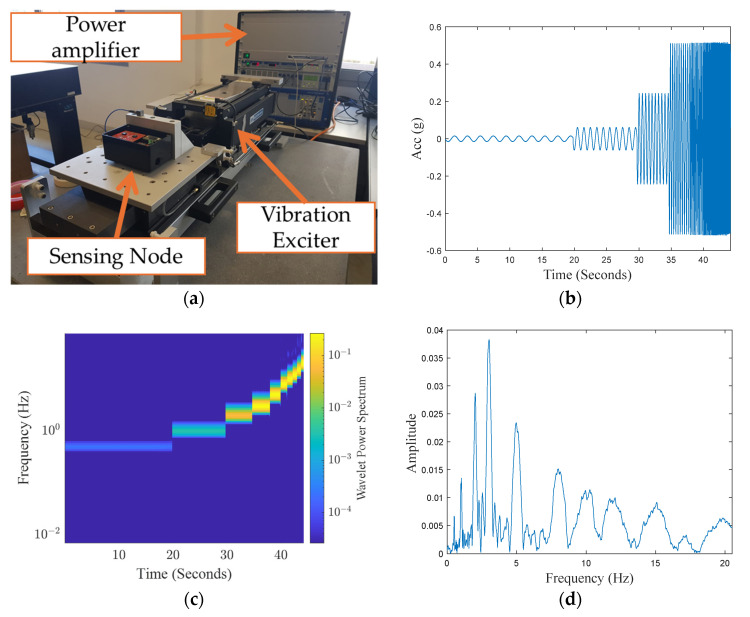
Vibration exciter test. (**a**) Setup adopted during the test along the *X*-axis. (**b**) Time series of the concatenated time window of 10 periods for each frequency [17]. (**c**) Wavelet analysis of the corresponding time window [17]; (**d**) Discrete Fourier transform of the concatenated signal.

**Figure 10 sensors-24-04023-f010:**
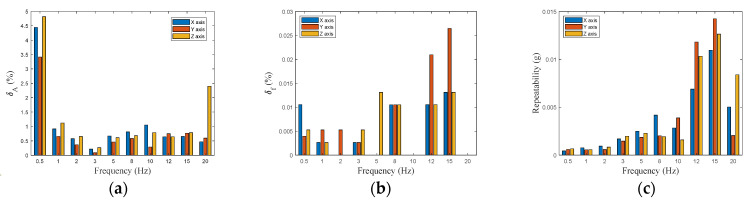
Vibration exciter test. Bar chart of indexes value under different operating conditions for each axis: (**a**) *δ_A_*, (**b**) *δ_f_*, (**c**) repeatability.

**Figure 11 sensors-24-04023-f011:**
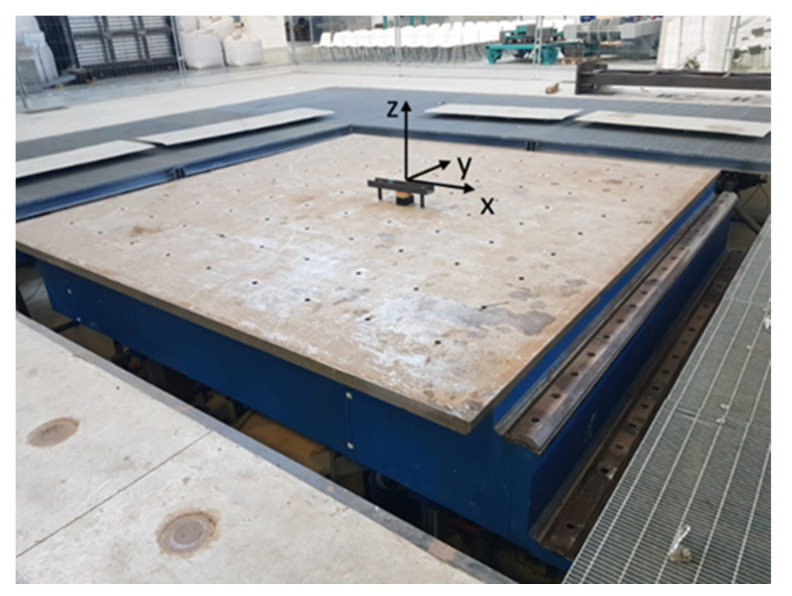
Setup employed during the tests with the vibrating platform that reports the axis orientation of the sensing system.

**Figure 12 sensors-24-04023-f012:**
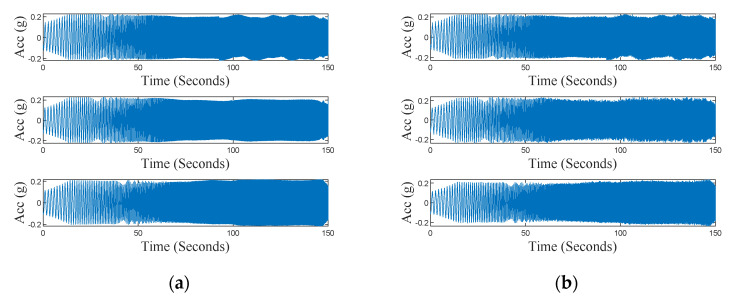
Time series along the *X*-axis (**top**), *Y*-axis (**center**), and *Z*-axis (**bottom**), in the frequency sweep test, recorded by (**a**) reference instrumentation; (**b**) the sensing platform.

**Figure 13 sensors-24-04023-f013:**
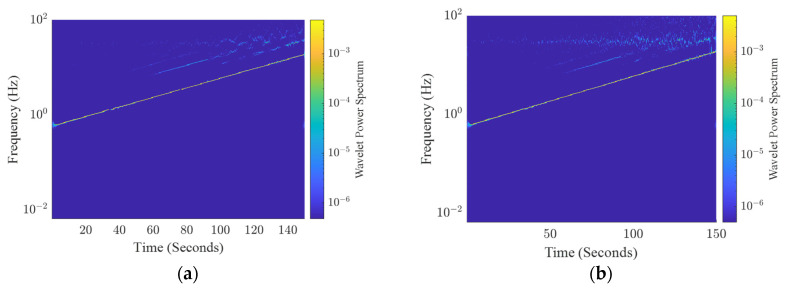
Wavelet analysis for the frequency sweep test. (**a**) Wavelet power spectrum of the reference instrumentation signals; (**b**) wavelet power spectrum of the sensing platform signals.

**Figure 14 sensors-24-04023-f014:**
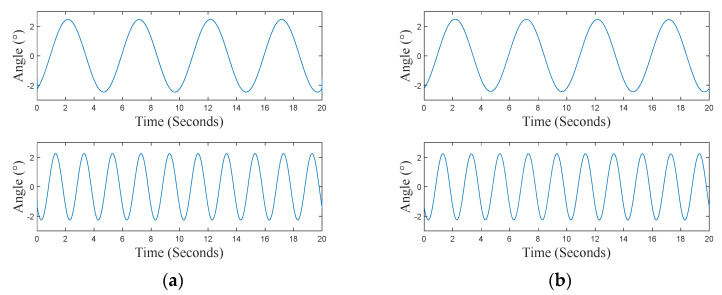
Time series recorded during the tilt test, in the case of the node rotation applied around the *Y*-axis. **Top**: 0.2 Hz; **bottom**: 0.5 Hz. (**a**) Reference signals; (**b**) sensing platform signals.

**Figure 15 sensors-24-04023-f015:**
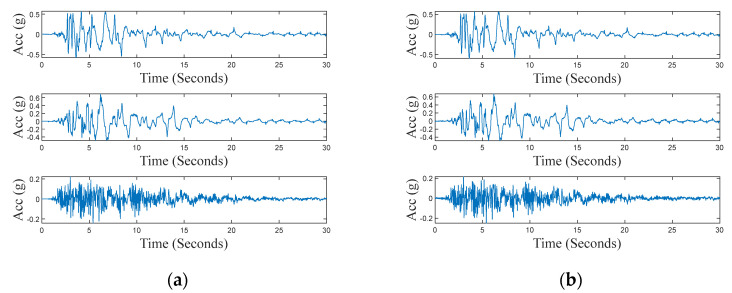
Time series of the seismic test along the *X*-axis (**top**), *Y*-axis (**center**), and *Z*-axis (**bottom**), recorded by (**a**) reference instrumentation; (**b**) the sensing platform.

**Figure 16 sensors-24-04023-f016:**
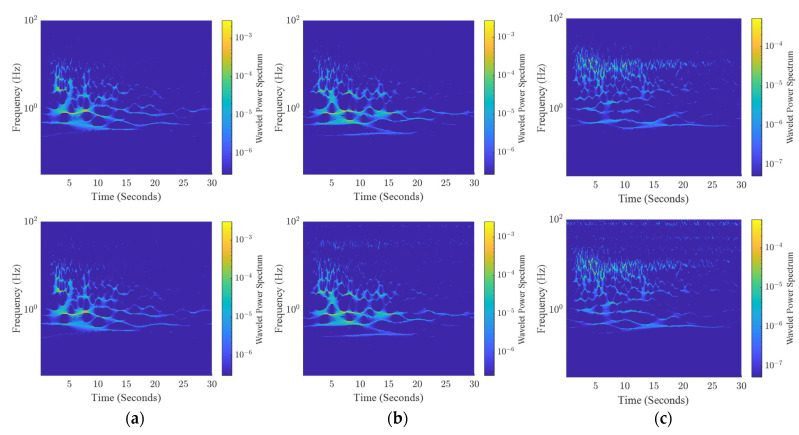
Wavelet power spectrum of the seismic test recorded by reference instrumentation (**top**) and the sensing platform (**bottom**), along: (**a**) *X*-axis, (**b**) *Y*-axis; (**c**) *Z*-axis.

**Figure 17 sensors-24-04023-f017:**
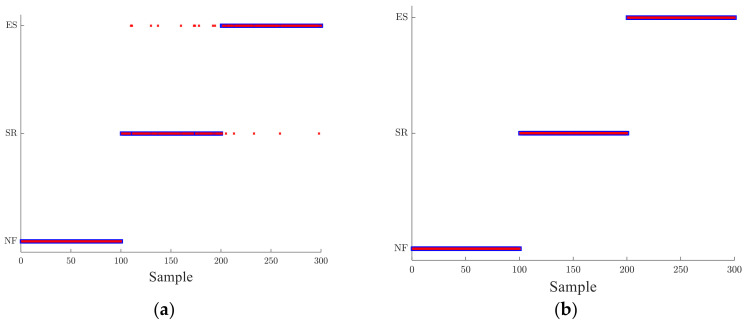
Validation test: expected data (blue symbols) and estimated data (red symbols). (**a**) Algorithm based on RMS values of the raw acceleration module; (**b**) algorithm shown in Figure 8 and exploiting rules (4).

**Table 1 sensors-24-04023-t001:** Number of patterns for each class.

	NF	ES	SR
Number of patterns	500	447	432

**Table 2 sensors-24-04023-t002:** Results of the resolution analysis for acceleration and tilt measurements [16].

	X	Y	Z
**Acceleration (mg)**	0.30	0.30	0.55
**Tilt (°)**	0.016	0.016	0.020

**Table 3 sensors-24-04023-t003:** Nominal value of frequency, *f*, and the amplitude, *A*, of applied signals.

***f* (Hz)**	0.5	1.0	2.0	3.0	5.0	8.0	10.0	12.0	15.0	20.0
***A* (g)**	0.015	0.060	0.241	0.510	0.510	0.510	0.509	0.510	0.510	0.510

**Table 4 sensors-24-04023-t004:** Performance indexes estimated for the frequency sweep test.

	*X*-Axis	*Y*-Axis	*Z*-Axis
** *ζ* **	2.76%	9.99%	12.86%
** *R_norm_* **	0.99	0.99	0.99
** *C_norm_* **	0.97	0.92	0.92

**Table 5 sensors-24-04023-t005:** Performance indexes estimated for the tilt test.

	0.2 Hz		0.5 Hz	
	*Y*-Axis	*X*-Axis	*Y*-Axis	*X*-Axis
** *ζ* **	1.66%	1.14%	2.15%	3.85%
** *R_norm_* **	0.99	0.99	0.99	0.99
** *C_norm_* **	0.99	0.99	0.99	0.94

**Table 6 sensors-24-04023-t006:** Performance indexes estimated for the seismic test.

	*X*-Axis	*Y*-Axis	*Z*-Axis
** *ζ* **	3.77%	5.05%	19.45%
** *R_norm_* **	0.99	0.99	0.98
** *C_norm_* **	0.98	0.97	0.84

**Table 7 sensors-24-04023-t007:** Confusion Matrix obtained for the classification algorithm shown in Figure 8 for the setting and test datasets.

			Classification Output		
			NF	SR	ES
**Expected Output**	**Setting dataset**	**NF**	400	0	0
		**SR**	0	400	0
		**ES**	0	0	400
	**Test dataset**	**NF**	100	0	0
		**SR**	0	100	0
		**ES**	0	0	100

## Data Availability

The data presented in this study are available: https://doi.org/10.6084/m9.figshare.26065126.

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
