# Peer review of "A Low-Cost Sensing Solution for SHM, Exploiting a Dedicated Approach for Signal Recognition"

_sensors, 2024, doi:10.3390/s24124023_

Round 1

Reviewer 1 Report

Comments and Suggestions for Authors

This paper presents the development of a low-cost sensing platform for use in the field of SHM, as well as a signal processing approach. Here are some comments:

Major comments:

(1)   This paper presents a low-cost sensor platform. What specifically do the authors consider to be low cost?

(2)   The horizontal coordinate of a general ROC curve is specificity and the vertical coordinate is sensitivity. Why are the horizontal coordinates of Figures 5 and 7 of the manuscript thresholds? Why can the threshold be estimated based on the above pictures? Please explain.

(3)   Is the title "exploiting a novel approach for signals recognition " appropriate? Because both CWT and ROC theory are established methods, although the authors use them well.

(4)    Line 214 of the manuscript should read to satisfy condition (1) instead of (3).

(5)   Manuscript equation (3) has the wrong numerator and should be TN.

(6)   Manuscript lines 198-199, 231 were incorrectly entered; RS should be SR.

Minor comments:

(1)   The text in Manuscript Figure 1 is blurry. It is recommended to improve the clarity of the image.

Comments on the Quality of English Language

some sentences should be further polished.

Reviewer 2 Report

Comments and Suggestions for Authors

This paper presents a novel approach to classifying vibration signals into three classes: Noise Floor, Exogenous, and Structural Response. Furthermore, it introduces a low-cost device capable of recording motion signals. The research is well-designed and executed, and the methodology is well-written but needs some clarifications.

The drawings and figures are clear to the reader, but some minor improvements are recommended to the authors. Providing a concise overview of the system's components and dimensions in Figure 2 and Figure 9(a) would be beneficial.

The manuscript has some minor typos, such as complaint instead of compliant in line 179. Please carefully check and revise the manuscript.

Please elaborate more on the variables of equations 1 and 7.

Please provide additional details regarding the "Shifting time check" step, as outlined in the flowchart (Figure 8).

The authors are encouraged to provide an analysis of the factors contributing to the substantial variance in the ζ values between the X-axis and Y-axis, as indicated in Table 4.

Does the Confusion Matrix depicted in Table 7 refer to the filtered data? If that's the case, authors should clarify it in the captions to increase the clarity of the manuscript. 

Comments on the Quality of English Language

In my opinion, the quality of the English language used in this manuscript is sufficient.

Please carefully read and revise the manuscript to identify and correct any minor typos.

Round 2

Reviewer 1 Report

Comments and Suggestions for Authors

no additional comments.